# The Dual-Edged Sword: Risks and Benefits of JAK Inhibitors in Infections

**DOI:** 10.3390/pathogens14040324

**Published:** 2025-03-27

**Authors:** Anders Jarneborn, Pradeep Kumar Kopparapu, Tao Jin

**Affiliations:** 1Department of Rheumatology and Inflammation Research, Institute of Medicine, Sahlgrenska Academy, University of Gothenburg, 413 46 Gothenburg, Sweden; pradeep.kopparapu@gu.se (P.K.K.); tao.jin@rheuma.gu.se (T.J.); 2Department of Rheumatology, Sahlgrenska University Hospital, 413 46 Gothenburg, Sweden

**Keywords:** Janus kinase inhibitors (JAKis), immunomodulation, autoimmune diseases, infection susceptibility, cytokine storm

## Abstract

Janus kinase inhibitors (JAKis) represent a relatively new class of immunomodulatory drugs with potent effects on various cytokine signalling pathways. They have revolutionized the treatment landscape for autoimmune diseases such as rheumatoid arthritis, psoriatic arthritis, and ulcerative colitis. However, their ability to modulate immune responses presents a dual-edged nature, influencing both protective immunity and pathological inflammation. This review explores the complex role of JAKis in infectious settings, highlighting both beneficial and detrimental effects. On the one hand, experimental models suggest that JAK inhibition can impair host defence mechanisms, increasing susceptibility to certain bacterial and viral infections. For example, tofacitinib-treated mice exhibited more severe joint erosions in *Staphylococcus aureus* (*S. aureus*) septic arthritis and showed impaired viral clearance in herpes simplex encephalitis. Additionally, clinical data confirm an increased risk of herpes zoster in patients receiving JAKis, underscoring the need for rigorous monitoring. On the other hand, JAK inhibition has demonstrated protective effects in certain infectious and hyperinflammatory conditions. In sepsis models, including cecal ligation and puncture (CLP) and *S. aureus* bacteraemia, tofacitinib improved survival by attenuating excessive inflammation. Furthermore, JAKis, particularly baricitinib, have shown substantial efficacy in mitigating cytokine storms during severe COVID-19 infections, leading to improved clinical outcomes and reduced mortality. These observations suggest that JAKis have a role in modulating hyperinflammatory responses in select infectious contexts. In conclusion, JAKis present a complex interplay between immunosuppression and immunomodulation. While they increase the risk of certain infections, they also show potential in managing hyperinflammatory conditions such as cytokine storms. The key challenge is determining which patients and situations benefit most from JAKis while minimizing risks, requiring a careful and personalized treatment approach.

## 1. Introduction

The immune system is a complex machinery, managing cellular stress, mutations, and constant exposure to pathogens. Under normal circumstances it operates seamlessly, ensuring the body’s processes run smoothly. However, when a pathogen is too cunning or completely unknown to the host, the immune response might struggle, become overwhelmed, or overreact, causing harm to the host [1]. It can even start to act out without a clear external threat, as seen in the multitude of conditions that we call autoimmune or autoinflammatory diseases [2]. Just a couple of decades ago, a person with rheumatoid arthritis might have had to endure a great reduction in quality of life, or in some serious situations, such as certain vasculitis diseases, had a prognosis and life expectancy comparable to that of malignancy [3,4]. Today, we have more effective treatment, though we sometimes lack a complete understanding of how they work, or which patient will respond best to a particular drug [3]. This uncertainty in details is also the case of the possible adverse effect of manipulating the immune system and the risks involving the diminished capacity to ward off external threats. Interplay between host and pathogen is an intricate game for dominance and evasion [5]. Interfering with this balance can have unpredictable effects and requires a cautious and humble approach to the potential consequences of any manipulation.

This review focuses on the effect of Janus kinase inhibitors (JAKis), a relatively new class of immune modulatory drugs, in the context of infections. We will explore how these drugs can yield both beneficial and detrimental outcomes depending on the clinical setting, highlighting the dual-edged nature of their impact on host immunity.

## 2. Janus Kinase and Its Inhibition

Research into the understanding of interferon-signalling led to the discovery of a new family of kinase, the Janus kinase (JAK), in the late 1980s and early 1990s [6]. While the essential functions of these kinases were subsequently identified, the complexity of these pathways, given the numerous components involved, leaves many questions unanswered [7]. A vast number of these pathways are involved in immune signalling, which soon made JAK an attractive target for the treatment of immunological disease. The first drug developed for this purpose was tofacitinib (brand name Xeljans), which was approved by the Food and Drug Administration (FDA) in 2012 for the treatment of rheumatoid arthritis. Since then, it has been approved for ulcerative colitis, ankylosing spondylitis, psoriatic arthritis, and juvenile idiopathic arthritis, with real-world data and experience indicating robust efficacy, convenient formulation, and rapid onset of action compared to many traditional drugs available [8]. The systemic JAKis are all administered orally, as tablets or solution, compared to many alternative drugs that require infusion or injection. Topical administrations are also available for skin diseases.

### 2.1. JAK: Mechanism of Action

Janus kinase is an intra-cellular component of activation associated with the receptors of over 50 different cytokines and hormones (a selection is shown in Table 1). The JAK is composed of several domains: a Four-point-one, Ezrin, Radixin, Moesin (FERM)-domain anchoring the complex to a receptor close to the cells’ plasma membrane, a Src homology 2 (SH2)-domain functioning as an adaptor for binding target proteins, a pseudo kinase domain, and the actual kinase domain, which has an adenosine triphosphate (ATP)-binding site for phosphorylation [9]. Upon binding of a cytokine to the extracellular binding site, the kinase part becomes activated and phosphorylates the SH2-domain. The phosphorylated SH2 acts as a binding site for cytoplasmic proteins, the main target being the signal transducer and activator of transcription (STAT) proteins. Binding of these to the phosphorylated SH2 domain enables the JAK to, in turn, phosphorylate the STAT proteins. In mammals, seven unique STATS have been identified [10]. Upon phosphorylation, these STATs will dimerize into homodimers or heterodimers and translocate to the nucleus to facilitate the transcription of various genes (Figure 1).

The body of evidence supporting the diverse effects of JAK-STAT signalling is steadily growing. Given their direct involvement in cytokine signalling, the JAKs are essential in immune system development and signalling, with JAK2 being more involved in haematopoiesis than the other three [11]. This is highlighted by the various JAK mutations identified in humans. For example, a mutation leading to a deficiency of JAK3 results in severe combined immunodeficiency in humans, characterized by severely impacted T- and NK-cell development and impaired B-cell function [12]. A rare mutation leading to TYK2 deficiency was found in a patient with atopic dermatitis with elevated IgE levels as well as increased susceptibility to viral, bacterial, and mycobacterial infections [13]. Mutations giving rise to JAK1 and JAK2 deficiency have not been identified in humans but it leads to perinatal and embryonic lethality, respectively, in mice. Gain-of-function (GOF) mutations in JAK1 and JAK3 are associated with leukaemia and lymphoma [7], while GOF mutations in JAK2 are highly associated with myeloproliferative disease including polycythemia vera, essential thrombocytopenia, and primary myelofibrosis [11].

Each JAK-dependant receptor has more than one subunit, each associated with a JAK. These can be identical or form different combinations (Table 1). JAK1, together with JAK2 or TYK2, is associated with type I and II interferon signalling, which is crucial for viral defence and antitumoral effects [14,15]. It also associates with the gp130 receptor subunit, whose ligand includes IL-6, a key cytokine in a pro-inflammatory response including effects on both myeloid and lymphoid cells [16]. TYK2 is associated with IFN-ɣ and the IL-12/IL-23 axis, which influence Th1 and Th17 differentiation, involved in the defence against both bacterial and fungal infections [17]. JAK2 is heavily associated with haematopoietic functions, being the sole JAK for erythropoietin (EPO), granulocyte-colony stimulating factor (G-CSF), and granulocyte macrophage–colony stimulating factor (GM-CSF) signal [18,19]. The common gamma chain exclusively associated with JAK3 and this subunit is utilized for the signalling of cytokines IL-2, IL-4, IL-7, IL-9, IL-15, and IL-21, which are all involved in lymphocyte activity, most dominantly T-cell activation, differentiation, and survival, but also important for NK-cells and innate lymphoid cells [20].

Natural negative regulation of JAKs include the suppressor of cytokine signalling (SOCS) proteins. These are a family of intracellular proteins that can bind the kinases, including JAKs, and compete with the binding of the STATs, thus blocking downward propagation of the signal, as well as leading to ubiquitination of its target. SOCS are induced by various cytokines as negative feedback, for example SOCS3 is induced by IL-6 and SOCS1 by IFNs, IL-12 and IL-2 [7]. Other components for negative regulation include protein inhibitors of activated STATs (PIASs), which prevent STATs from forming dimers or direct dephosphorylation of STATs by protein tyrosine phosphatases [11].

Several cytokines share the same JAKs and STATs, and the outcome of the cytokine signal can still be diverse, indicating the complex network of this signalling (Table 1). Various mechanisms for the differentiated effect, despite the limited number of STATs involved, include synergy with other receptor pathways, effects on histones and epigenetic qualities, specific STAT interactions of antagonism or synergy, and combinations of STATs with other transcription factors for specific outcomes [7].

**Table 1 pathogens-14-00324-t001:** Selection of cytokines utilizing the JAK/STAT pathway for signalling.

Cytokine	JAKs	Primary STATs	Main Function
Il-2	JAK1, JAK3	STAT5A/B, STAT3, STAT1	T-cell proliferation, NK-cell boosting [21].
IL-4	JAK1, JAK3	STAT6	Th2 differentiation [22].
IL-6	JAK1, JAK2, TYK2	STAT1, STAT3	Acute phase response, myeloid cell stimulation, lymphoid differentiation [16].
IL-7	JAK1, JAK3	STAT5, STAT3, STAT1	Lymphocyte survival and proliferation [23].
IL-10	JAK1, TYK2	STAT3	Anti-inflammatory through inhibition of monocytes, DCs, inhibition of proinflammatory cytokine production [24].
IL-12	JAK2, TYK2	STAT4	Promotes Th1 response in Th-cells. Stimulates production of IFN-ɣ. [25]
IL-15	JAK1, JAK3	STAT3, STAT5	Stimulates T-cell and NK-cell response [26].
IL-21	JAK1, JAK3	STAT3, STAT1, STAT5	Th-cell differentiation, activity of follicular Th-cells, B-cell regulation [27].
IL-23	JAK2, TYK2	STAT3, STAT4	Promotes Th17 differentiation [25].
GM-CSF	JAK2	STAT5A/B	Myeloid cell proliferation, macrophage maturation [18].
EPO	JAK2	STAT5	Stimulation of erythropoiesis [19]
GH	JAK2	STAT1, STAT3, STAT5	Stimulates bone and muscle growth, affects fat and protein metabolism, glucose homeostasis [28].
TPO	JAK2	STAT5, STAT3	Haematopoietic stem cell survival, megakaryocyte proliferation [29].
IFN-α IFN-β	JAK1, TYK2	STAT1, STAT2	Antiviral, promotes antigen presentation, antiproliferative [14].
IFN-ɣ	JAK1, JAK2	STAT1, STAT3, STAT5	Antiviral, antitumoral, stimulates T-cells and NK-cells, primes macrophages [15].

### 2.2. Janus Kinase Inhibitors

The first generation of JAK inhibitors work by binding to the ATP domain of JAKs, thereby preventing the phosphorylating action of the receptor [30]. This binding is reversible and relatively fast-acting. Different JAKis have different affinity for the JAKs with varying degrees of specificity, which is often dose-dependent and may involve some overlap. A second generation of JAKis aims for greater selectivity, and some feature non-reversible binding [31]. Additionally, some compounds target other part of the receptor than the ATP-binding site, such as deucravacitinib, an allosteric inhibitor targeting the psudokinase domain of TYK2 [32]. Some effects may therefore be specific for different compounds, while others appear to be class effects common to all JAKis, such as an elevated risk for herpes zoster. Several different drugs are approved and in clinical use (Table 2).

The first JAKi approved for treatment of immune-mediated disease was tofacitinib, initially for rheumatoid arthritis. The drug showed superior effect versus placebo as monotherapy in patients who had an inadequate response to at least one previous disease-modifying antirheumatic drug (DMARD) [33]. It also showed superior efficacy as monotherapy compared to methotrexate (MTX) in patients who have not previously received therapeutic doses of MTX [34,35]. Additionally, tofacitinib in combination with MTX exhibited similar efficacy to anti-tumour necrosis factor (TNF) therapy with MTX in patients with inadequate response to MTX alone [36,37]. Although originally thought to be a selective inhibitor of JAK3, tofacitinib is now considered an inhibitor of all JAKs, with greater affinity for JAK 1 and JAK3, and to a lesser extent JAK2 [38].

Given its effect on several JAKs and the vast involvement of these kinases in both immunological and other cellular activities, pinpointing distinct mechanisms of action can be challenging, and the field is still in its early stages of exploration. Proposed key factors in the beneficial effect of tofacitinib in autoimmune disease include decreased synthesis of degrading matrix metalloproteases in the synovium [39], altered differentiation of Th-cells and inhibited interleukin (IL)-6 signalling [40], and modified activity of dendritic cells (DCs) [41].

An elevated risk for venous thromboembolism (VTE), major adverse cardiovascular events (MACE), and malignancy has been observed in early studies of tofacitinib as well as some other JAKis. These concerns prompted the FDA-mandated post-marketing ORAL Surveillance study [42]. This study, which assessed the risk of MACE and malignancy in patients with at least one cardiovascular risk factor, suggested an increased risk of MACE, malignancy, and VTE compared to TNF-inhibitors (TNFis). These results led to the addition of strong safety warnings and a subsequent adjustment in the use of tofacitinib.

These concerns were extrapolated to include all JAKis across all indications. While the future role of JAKis in the global treatment landscape remains to be fully defined, more real-world data are still needed before tofacitinib and the other JAKis can secure their definitive place in the treatment arsenal for rheumatic disease.

## 3. Dual Effects JAKi in Experimental Infection Models

### 3.1. Impact of JAKis in Infection Models

#### 3.1.1. JAK/STAT Pathway Mutants

Several studies have investigated the impact of the JAK/STAT pathway in animal models of infections, with the majority of studies evaluating the effect in different models of sepsis (Table 3). The JAK/STAT pathway is involved in multiple mechanisms critical to sepsis pathophysiology [43]. These include Th-cell differentiation and activation, where such key cytokines like interferons, IL-2, and IL-4 depend on JAKs for signalling [40]. The key anti-inflammatory cytokine IL-10 also signals through JAK1/TYK2 [24]. In addition, JAKs are involved in macrophage polarization [44].

Using cecal ligation and puncture (CLP), a model of polymicrobial sepsis and shock, STAT1-deficient mice had significantly higher survival and less hypothermia and less hepatic injury, while TYK2-defiecent mice had no significant advantage in survival compared to WT [45]. In a model of endotoxin-induced shock, STAT1-, STAT4-, and TYK2-defiencent mice all showed higher survival than WT [48], with a tendency towards lower levels of TNF-α, IL-12, and IFN-ɣ in serum of deficient mice. Similar findings were found in another study on lipopolysaccharide (LPS)-induced shock where mice lacking TYK2 showed complete protection from mortality, while a lack of STAT1 showed reduced but not significant protection [49]. However, in this study, STAT2−/− mice showed a significant increase in mortality, accompanied by decreased pro-inflammatory cytokines and increased expression of endothelial ICAM-1. The authors proposed a disruption of STAT2-dependent NF-kB signalling rather than a loss of type 1 interferon signalling as the cause of mortality, as Interferon-alpha/beta receptor alpha chain 1 (Ifnar1)-deficient mice were protected against shock. Furthermore, in yet another study, both STAT6−/− and STAT4−/− were shown to have higher mortality than WT [51]. IL-12 blockade rescued STAT4−/− mice from death indicating a possible STAT4-independent effect of this cytokine. STAT6−/− was associated with higher levels of pro-inflammatory mediators and a higher degree of leukocyte infiltration in the liver and lungs. However, STAT4−/− and STAT6−/− showed better survival in a mouse CPL model [47]. These two STATs are considered typical for a Th1 and Th2 response, respectively, with a corresponding role in IL-12 and IL-4 signalling. Specifically, STAT6−/− showed markedly better bacterial clearance measured as colony-forming units (CFU)/μL in both blood and peritoneal fluid. Levels of TNF-α and IL-12 were higher on the peritoneal fluid and blockade of these cytokines abrogated the elevated bacterial clearance. STAT4−/−, on the other hand, showed no difference in bacterial clearance, but less signs of systemic organ damage, suggesting different roles in their protective effects, despite their opposing roles in Th-cell differentiation. STAT3, a key in IL-10 signalling, is critical in dampening and resolving inflammation. Matsukawa et al. showed that STAT3-deficient mice had worse survival than WT mice in a CPL model, with higher levels of pro-inflammatory cytokines and greater organ damage measured by serum markers and histopathology [46]. The same pattern was seen with LPS-challenged mice with targeted disruption of STAT3 in neutrophils and macrophages [50].

#### 3.1.2. JAK/STAT Pathway Inhibition in Infection Models

Inhibition of the JAK/STAT pathway has also been the target of treatment studies in animal models. Studies of JAKis showing underlaying mechanisms of heightened risk of infection are relatively scarce. The absolute majority are concerned with manipulating the immune system for a positive outcome, and sepsis is the usual setting for these models (Table 4). Several animal models have been utilized to study the effect of JAKi treatment in bacterial, viral, and fungal infections. For bacterial infections, the well-established cecal ligation and puncture (CLP) model closely mimics the clinical presentation of polymicrobial, Gram-negative sepsis, as seen in appendicitis or diverticulitis with tissue ischemia [52]. The *S. aureus* septic arthritis and sepsis model replicates hematogenous septic arthritis and sepsis caused by Gram-positive *S. aureus* [53]. For viral infections, the HSV-1 encephalitis mouse model has been used to study both latent and primary HSV-1 encephalitis [54]. Furthermore, the murine model of hematogenous disseminated candidiasis has been employed to investigate the effects of JAK inhibitors in candida sepsis [55]. There are also toxin-induced shock models that better mimic hyperinflammatory responses triggered by bacterial components without live bacteria. These include models using lipopolysaccharide (LPS) from Gram-negative bacteria [56] or a combination of *S. aureus* enterotoxin and LPS [57], both of which induce severe systemic inflammation and cytokine storm–like responses. These models are particularly useful for studying the pathophysiology of septic shock and evaluating potential therapeutic interventions targeting excessive inflammation.

Hui et al. used the compound AG490 known to inhibit JAK2, which inhibits STAT3 activation, in a rat model of CLP [58]. A subcutaneous injection of AG490 (8.0 mg/kg) 30 min before CLP significantly improved 48 h survival compared to untreated mice, along with reduced tissue damage in the lungs and liver. AG490 was also evaluated by Peña et al. in a model of CLP and endotoxemia models [59]. An intraperitoneal injection of AG490 (5 mg/kg–10 mg/kg) administered 30 min before LPS challenge or 12 h after CLP significantly improved survival compared to the control mice in both models. Results further indicated that JAK2 inhibition by AG490 limited activation of NF-kB, dampening the inflammatory response.

Tsirigotis et al. investigated the effect of JAK1/JAK2 inhibitor ruxolitinib at different doses and time points in a mouse model of candida sepsis [63]. The highest oral gavage dose (50 mg/kg/day), started one day before infection, showed the lowest histopathological score of inflammation, but the highest fungal load. Untreated controls had the highest degree of inflammation and lowest fungal load. The high-dose group had a significantly worse survival than the controls. A medium dose of 6.25 mg/kg, also started one day before infection, showed intermediate levels of inflammation and fungal load, but had a significantly higher survival than both the high dose and the controls. Intraperitoneal injection of ruxolitinib (0.67 mg/kg) 30 min before LPS challenge also improved survival in a model of LPS-induced sepsis studied by Li et al. [61], with an accompanied reduction in circulating TNF-α, IL-6, IL-23, and IL-1β, as well as less kidney and liver damage on histopathological examination.

Tofacitinib has also been tested in a murine setting. Zhang et al. tested three doses of tofacitinib (1 mg/kg, 3 mg/kg, and 10 mg/kg) in a rat model of CLP [60]. Administered via oral gavage 2 h after CLP and every 6 h for 7 days, the highest dose resulted in significantly higher survival compared to the untreated controls, accompanied by less tissue pathology in the lungs and reduced levels of pro-inflammatory cytokines TNF-α, IL-1β, IL-6, and IFN-ɣ as measured at protein and mRNA levels in the lungs. Reduced levels of NF-kB in the lungs were seen in tofacitinib-treated mice, and the authors suggested that a JAK-STAT3 enhancing mechanism may contribute to the positive effect of tofacitinib in this model.

Yun et al. tested the effect of tofacitinib on kidney injury in a murine model of LPS-induced acute kidney injury [62]. Three doses of tofacitinib (5 mg/kg, 10 mg/kg, or 15 mg/kg) were administered by oral gavage every 6 h until sacrifice at 24 h, and tofacitinib dose-dependently reduced signs of kidney injury, as evaluated by serum markers and histopathological scores. Tofacitinib-treated mice also exhibited lower levels of cytokines TNF-α, IL-1β, IL-6, and IFN-ɣ, as measured by quantitative real-time polymerase chain reaction (qRT-PCR) and Western blot.

Singh et al. tested high doses (150 mg/kg or 300 mg/kg) of tofacitinib by oral gavage 6 h after onset of sepsis in a mouse CLP model [67]. Although survival was not monitored, histological examination showed a protective effect of 150 mg/kg, with less tissue injury in the lungs, liver, and kidneys compared to the untreated CLP group. However, the 300 mg/kg group showed increased signs of damage in liver and kidneys, and the authors speculated that this might be an adverse effect of the substantial dose.

Recently, we studied the effect of tofacitinib in *S. aureus* bacteraemia using superantigen SEA–producing *S. aureus* strain AB-1. At the given high dose of 1 × 10^7^ CFU/mL per mouse, many expected clinical features bear similarities to sepsis, with a change in body temperature, gastrointestinal symptoms, and high mortality. Given the known protective effect on Th-cells in murine sepsis, and the known pathogenic effect of IFN-ɣ and proinflammatory cytokine IL-6, both of which utilize JAKs for signalling, we hypothesized that tofacitinib could have a beneficial effect in this context. We showed that tofacitinib, administered via a subcutaneous (sc.) pump (15 mg/kg/day) starting 3 days before infection, even in the absence of antibiotics, significantly prolonged survival, while not rescuing the mice from succumbing to infection [64]. This result is in line with previous studies of JAK inhibition in other models of sepsis.

The effect of tofacitinib was further studied in a mouse model of toxin-induced shock [64]. The shock is induced by *S. aureus* superantigen toxin TSST-1. As mice are more resistant to superantigens than humans, a second potentiating agent is required. For this purpose, mice are given an ip. injection of LPS 4 h after the initial toxin challenge. Pre-treatment with tofacitinib had a strong protective effect in this model, rescuing 70% of mice when given via subcutaneous (sc.) pump at 15 mg/kg/day, and 100% of animals when given as two sc. injections of 50 mg/kg. Surviving mice appeared to make a full clinical recovery, while control mice receiving the vehicle only succumbed to shock within 24 h. At 24 h after the first toxin injection, serum levels of TNF-α and IFN-ɣ were markedly reduced in tofacitinib-treated mice, suggesting a mitigated inflammatory response, consistent with findings from several of the above-mentioned studies on JAK inhibition and STAT deficiency. Tofacitinib also prevented the hypothermia seen in mice succumbing from shock. However, treatment after the toxin challenge did not have any beneficial effect.

T-cell activation is an essential mechanism behind superantigen-induced shock. This is supported by the known mechanism of the superantigen activity, and by previous studies showing that cyclosporin A, a known T-cell–suppressing drug, protects animals from death in a model of *S. aureus*–derived superantigen TSST-1 and SEB plus LPS, while having no effect in shock induced by LPS alone [68,69]. The superantigen-induced toxic shock severely reduces the numbers of circulating T-cells, but this loss was not dampened by tofacitinib [65]. However, activation of the circulating cells was significantly higher in the untreated mice, suggesting that tofacitinib inhibits T-cell activation in the setting of toxic shock. To further characterize the cytokine environment influencing differentiation and activation of Th-cells, circulating levels of Th2 cytokine were significantly lower in toxic-challenged mice, with no difference between tofacitinib-treated mice or the controls. The mRNA expression of prototypical Th1 cytokine IFN-ɣ, as well as another key Th1 cytokine, IL-12, was significantly higher in untreated mice, compared to the healthy controls. Tofacitinib mice, however, did not differ significantly from the controls, indicating that tofacitinib leads to a less Th1-promoting environment, which might explain part of its protective effect. The importance of IFN-ɣ is also highlighted by the finding that blocking this cytokine by pre-treatment with neutralizing antibodies can protect mice from TSST-1-LPS–induced shock [69].

### 3.2. Detrimental Effects of JAKi in Infection Models

A recent publication demonstrated that treatment with tofacitinib via subcutaneous (sc.) pump at 15 mg/kg/day 3 days before infection has a negative impact on septic arthritis caused by *S. aureus* [64]. Mice treated with the drug and subsequently inoculated with a non-lethal arthritogenic dose of bacteria showed a similar degree of clinically judged arthritis over the course of 10 days. The bacterial load in the kidneys of mice sacrificed on day 10 as well as an observable kidney abscess, is used as a marker of the ability to systemically clear the bacteria from the body. No difference was seen between tofacitinib-treated mice and the controls. This was also the case of weight development, another way to assess the degree of systemic infection in this model, and no difference was seen between groups. When joints of mice sacrificed on day 10 were scanned using a μ-computer tomography, a significant difference revealed itself between the two groups. Mice treated with tofacitinib showed a significantly higher degree of bone erosion than the controls. Cytokine levels on day 10 showed higher levels of IL-6 in tofacitinib compared to the controls, but no difference in TNF-α, IFN-ɣ, monocyte chemoattractant protein-1 (MCP-1), or receptor activator of nuclear factor kappa-Β ligand (RANKL) measured in the serum. Experience from the septic arthritis model has shown that levels of IFN-ɣ generally are low towards the end of the course of infection on day 10. It has previously been shown that mice deficient in IFN-ɣ receptor develop more arthritis, indicating a protective role of this cytokine [70]. Lower levels in tofacitinib-treated mice could account for worsened outcome in the joints, but this is still speculative until confirmed with further experiments.

The effect of tofacitinib treatment in primary and latent herpes simplex encephalitis was studied in a mouse model [66]. In primary infection, treatment with tofacitinib via a subcutaneous (sc.) pump at 15 mg/kg/day, starting 48 h before HSV-1 exposure, led to significantly worse clinical symptoms and weight loss. The viral load in both brain and trigeminal ganglia were higher in tofacitinib-treated animals, indicating a clear debilitating effect of tofacitinib on the response to the virus. The immune response to HSV-1 consists of both more specific antiviral components and general pro-inflammatory mediators. Analysis of mRNA expression of key cytokines and chemokines showed that in the brains of mice treated with tofacitinib, key antiviral components IFN-α, IFN-ɣ, and CXCL10 were downregulated compared to the control mice. Conversely, more general pro-inflammatory mediators IL-1β, TNF-α, and chemokine CXCL1 were all upregulated in tofacitinib-treated mice compared to the controls, indicating a disruption of the specific antiviral response. Flow cytometry analysis of cells in infected brain tissue revealed little difference in NK-cells, CD4+ T-cells, microglia, or monocytes, while CD8+ T-cells were more abundant in tofacitinib-treated mice. However, further characterization of cells important to the early response to HSV-1 CNS infection, microglia, and infiltrating monocytes showed a difference in polarization patterns. Both these cell types can differentiate towards a more pro-inflammatory or anti-inflammatory phenotype: M1 and M2. Tofacitinib-treated mice had a lower ratio of M1 monocytes, and higher amounts of M2 among both monocytes and microglia. Further in vitro analysis showed that HSV-1 promotes a shift towards M2, and that tofacitinib seemed to further amplify this shift in peritoneal macrophages. mRNA expression of chemokines CXCL9 and CXCL10 and IFN-α was blocked by tofacitinib regardless of macrophage phenotype. In cultured microglia and monocytes that were stimulated in vitro with HSV-1 or poly I:C, a compound that mimics viral DNA, tofacitinib had no effect on IFN-α expression, but significantly decreased both CXCL9 and -10 in both cell types. In summary, tofacitinib has a detrimental effect on primary herpes simplex encephalitis in mice, with aggravated symptoms and a higher viral load in the brain. We propose that in addition to the ability of tofacitinib to inhibit the signalling of key cytokines (i.e., interferon type 1 and 2, both of which rely on JAKs for signalling, though not specifically studied in these experiments), tofacitinib also impacts microglial and monocyte differentiation towards a less specific antiviral response, resulting in more viral replication and more damage from unspecific pro-inflammatory mediators, such as TNF-α and IL-1β.

Regarding latent infection, the administration of tofacitinib was limited in time and no clinical signs of reactivation were observed. However, viral titres in treated mice were higher in tofacitinib-treated mice. While no real conclusions can be drawn from these limited results, it could warrant further study through a different design. Low-grade, subclinical inflammation caused by viral activity could have long-term negative effects, being implicated in neurodegenerative disease [71]. Whether the higher viral titres seen in tofacitinib-treated mice give a sub-clinical, low-grade inflammatory response could be of interest in further studies.

## 4. JAK Inhibitors and Human Infectious Disease

Increased risk of infection is associated with all approved JAK inhibitors with the risk being comparable across all agents, regardless of target. What really distinguishes this class of drugs from other comparable groups (i.e., the biologics) is the marked increase in herpes zoster cases, observed with most members of the class. Since the approval of the first JAKi, ruxolitinib, in 2011, real-world experience has accumulated over the years, providing a fair amount of data to support the various associated risks. In all studies mentioned below, the JAKi is administered orally, except delgotinib, which is a topical formulation for the treatment of local skin disease.

### 4.1. Tofacitinib (Xeljans)

Tofacitinib was the JAKi first approved for an immune-mediated inflammatory diseases (IMIDs), specifically for RA. The majority of data on infection risk comes from studies on patents with RA treated with tofacitinib. Overall, tofacitinib is generally considered safe in terms of infections, although there is a recognized, possibly elevated risk that requires vigilance from both patients and health care workers [72]. Of note, the only dose approved for clinical use in RA is 5 mg twice daily (bid).

Early randomized controlled studies (RCTs) comparing tofacitinib versus placebo or methotrexate indicated a trend towards an increased risk of serious infections in the group treated with 10 mg tofacitinib daily [33,34]. However, actual numbers of cases were low, and the drug was generally considered safe with regards to infection.

One infection that stood out throughout the clinical program for tofacitinib was the reactivation of varicella virus, also known as herpes zoster (HZ). This is not unexpected, as inhibition of JAK affects IFN-signalling known to be key in antiviral defence [73]. Similarly, an elevated risk of HZ has been observed among biologics, particularly with monoclonal antibody anifrolumab, which targets type 1 IFN receptor subunit 1 [74]. Pooled data from the phase II, III, and long-term extension (LTE) studies showed an incidence rate of HZ of 4.4 (95% CI 3.8, 4.9) for tofacitinib. Local differences were observed, with Japan and Korea showing a higher incidence rate of 9.2 (95% 7.5, 11.4). Real-world US registry data confirmed that this elevated risk of HZ was prominent for tofacitinib compared to other biologics [75]. The hazard ratio (HR), adjusted for factors such as age, gender, and glucocorticoid use, was 2.01 (95% 1.40, 2.88) compared to a reference biologic, abatacept. No other biologics available to treat RA at the time, including five different TNFi, rituximab, and tocilizumab, showed a significant different HR compared to abatacept. Overall, the risk of HZ is a clear and well-documented concern in patients treated with tofacitinib. However, serious disease or dissemination appears rare, and vaccination seems to offer protection in this setting [73].

Over time, accumulated data has given a more reliable understanding of the infectious risks. A pooled analysis of safety data from phase I, II, III, and LTE studies over 9.5 years included 7061 patients (total exposure of 22,875 patient-years (PY), with median exposure of 3.1 years). This study showed an incidence rate (IR) of 2.5 (95% CI: 2.3, 2.7) for serious infections, and 3.6 (3.4, 3.9) for herpes zoster [76]. The most common serious infections were pneumonia, herpes zoster, urinary tract infections, and cellulitis. Tofacitinib treatment included both 5 mg and 10 mg bid, as monotherapy and in combination with other csDMARDs, mainly MTX. The infection rate declined over time. A meta-analysis of randomized controlled phase II and III studies from the tofacitinib clinical programs included 5888 patients given tofacitinib 5 mg bid or placebo [77]. IR for serious infections was similar to other studies, at 1.97 (95% CI: 1.41, 2.68) compared to 1.19 (95% CI: 0.51, 2.34) in the placebo group, with a non-significant incidence rate ratio (IRR) of 1.22 (95% CI: 0.60, 2.45) compared to placebo.

Following approval, several studies assessed safety with data from real-world use [78]. Typically, tofacitinib is compared to other equivalent treatment options, such as biologics. One large registry study of US patients found an IR of 3.12 (95% 2.51, 3.84), slightly higher than the IR of pooled RCTs [79]. Compared to biologics (all types), no significant risk for total serious infections was noted. The risk for HZ was, however, significantly increased, with a 2-fold increase in risk in tofacitinib-treated patients compared to bDMARDs. A Swedish registry study compared adverse events in 20,117 RA patients starting a bDMARD or tsDMARD from 2010 to 2020 [80]. It also compared this group to sex- and age-matched controls in the general population. Overall, among treated RA patients, the hazard ratio of a serious infection was 2.68 (95% CI 2.56, 2.81) compared to the general population. Comparison between tofacitinib and TNFi etanercept, the largest treatment group used as a reference, was non-significant at 0.89 (95% CI 0.47, 1.69). However, for herpes zoster, tofacitinib had a significantly increased risk with a HR of 4.00 (95% 1.59, 10.06) compared to etanercept.

In 2022, the ORAL SURVEILLANCE study was presented [42]. This was a mandated post-approval phase IIIb/IV RCT focused on safety and adverse events observed in previous clinical studies. The study included patients over 50 years old with at least one cardiovascular risk factor. Patients were randomized to receive either tofacitinib 5 or 10 mg bid, or a TNFi (adalimumab in the US, etanercept in the rest of the world). Primary outcomes were focused on MACE and malignancy, although infections were reported as a secondary outcome. The hazard ratio for serious infection was significantly increased for tofacitinib 10 mg bid compared to TNFi. While the HR was numerically higher for tofacitinib 5 mg bid, the difference was not statistically significant. Both doses of tofacitinib showed a significantly increased risk of HZ. A post hoc analysis of infections in this study population revealed a significantly higher HR for non-serious infections in both tofacitinib groups [81]. However, no significant difference in serious infections was found between tofacitinib 5 mg bid and TNFi across any age category.

Overall combined data on tofacitinib is consistent, showing a slightly elevated but acceptable elevated risk of serious infections in patients treated for RA. The risk does not appear to be higher than other treatment options for active RA, except for herpes zoster infections that present a notably higher risk compared to biologic DMARDs (bDMARD).

Tofacitinib is also used to treat other conditions, though there is generally less data available compared to RA. Tofacitinib is used for treatment of inflammatory bowel disease. Like the case of RA, the longest follow-up study in the clinical program for tofacitinib in ulcerative colitis showed a dose-dependent increase in herpes zoster cases, with 0.5% of patients in the placebo group affected, compared to 1.5 and 5.1 in tofacitinib 5 mg and 10 mg, respectively. A study of real-world data in UC, using approved does of 10 mg bid for induction therapy, followed by 5 mg bid for maintenance of remission, reported an IR for HZ of 6.9 (95% CI 4.5, 9.3) [82]. In a systematic review from 2024, Chen et al. showed a similar pattern in inflammatory bowel disease (IBD) patients as seen in RA patients regarding infections with increased risk for tofacitinib compared to placebo [83]. Risk was elevated for both serious infections and herpes zoster in both ulcerative colitis and Crohn’s disease. However, the number of studies was relatively small, and neither difference was statistically significant. This population also tends to be younger and have fewer comorbidities than the RA population. For psoriatic arthritis, a long-term follow-up study in the clinical program, using a dose of 5 mg bid or 10 mg bid, found a similar pattern of elevated HZ rates, though possibly somewhat lower than for RA [84]. Similarly, a 48- week follow-up of tofacitinib 5 mg bid for treatment of ankylosing spondylitis showed 5 cases of HZ in patients receiving tofacitinib, compared to none in the placebo group [85]. Tofacitinib has also been used in children as young as two years old for treatment of juvenile idiopathic arthritis. A phase III withdrawal study of patients with juvenile idiopathic arthritis, using weight-based dosing of 5 mg bid or lower, found an estimated incidence rate per 100 PY of 2.4 (95% CI 0.5, 7.1) for serious infections and 1.6 (95% CI 0.2, 5.9) for herpes zoster [86]. In summary, the safety profile regarding infection seems consistent across different diagnoses. However, more data and experience are needed to determine whether any clear differences emerge across different conditions.

### 4.2. Comparisons of JAKis for Treatment of RA

The risk profile derived from clinical programs and limited real-world data for the whole group of JAKis in the treatment of RA is similar to that of tofacitinib. Following the revision of safety guidelines based on the above-mentioned ORAL Surveillance study, the recommendations for caution were extended to all JAKis. Figure 2 illustrates the approval of JAKis for treatment of RA.

In one meta-analysis of different JAKis for treatment of RA, baricitinib showed an estimated IRR for serious infection of 0.80 (95% CI 0.46, 1.38) compared to placebo [77]. None of the included JAKis (tofacitinib, baricitinib, and upadacitinib) showed a significant difference from placebo regarding serious infections. However, for HZ, baricitinib stood out, as it was the only JAKi showing a significant IRR of 2.86 (95% CI 1.26, 6.50) for HZ compared to placebo. While both other JAKis had a numerically higher IRR than placebo, neither reached statistical significance. This might be influenced by the fact that tofacitinib and upadacitinib studies included the lower doses of 5 mg and 15 mg, respectively, while baricitinib was used at the higher dose of 4 mg. A dose-dependent relationship between JAKis and HZ has been observed. However, the doses used in this study are all the approved, clinically used doses in Europe.

All in all, the pattern is similar across the different JAKis in treating the RA population, and any detailed direct comparison between different studies is not possible. In summary, an increase in risk of serious infection is seen, comparable to other treatment options, but an elevated risk of HZ stands out for the JAKis. The elevated risk of HZ seems to encompass the whole class of drugs.

### 4.3. Baricitinib (Olumiant)

Baricitinib is a JAK1 and JAK2 inhibitor approved for RA, AD, AA, and JIA [30]. The approved doses are 2 mg daily in the US by the FDA and 4 mg daily in Europe by the EMA, with the recommendation that a lower dose of 2 mg should be used in patients with old age, or those with risk factors for VTE, MACE, and recurrent infections.

Regarding serious infections, combined final results from phase I, II, III, and LTE of baricitinib included 3770 patients (14,744 PY), receiving 2 mg or 4 mg once daily, and were presented in 2022 [87]. IR for serious infections was 2.58 (95% CI 2.33, 2.86). The most common serious infections were pneumonia, HZ, urinary tract infection, and cellulitis. IR of herpes zoster was 3.0 (95% CI 2.70, 3.28). Of the 422 cases of HZ, 15 were considered serious (ophthalmic, meningitis, or multidermal).

A combined report for baricitinib in patients treated for AD included 2636 patients with a total exposure of 4628.4 patient-years using either 2 mg or 4 mg once daily [88]. Across all patients, an IR for serious infection was 1.8 (95% CI not stated). The most common were eczema herpeticum, cellulitis, erysipelas, and pneumonia. The IR for HZ was 2.8 and included multidermal or disseminated infections. Of note, this study also reported the incidence of herpes simplex infection, with total cases of HSV (IR 6.7). HSV reactivation is more common in individuals with AD [89]. Interestingly, few studies on patients treated for RA reported any incidence of HSV.

### 4.4. Upadacitinib (Rinvoq)

Upadacitinib is a JAK1 inhibitor approved for treatment of RA, PsA, AS, IBD, and AD [30]. Trials are ongoing for treatment of giant cell arteritis. For rheumatic diseases, the dose is 15 mg once daily, while for other diagnoses, doses range between 15 and 45 mg once daily.

Combined safety data from the clinical trial program for upadacitinib, including phase III and LTE data, involved patients receiving either upadacitinib 15 mg or 30 mg daily, with or without MTX [90]. Pooled results showed an estimated IR for serious infection of IR 3.1 (95% 2.5, 3.9) and 6.2 (95% 4.0, 6.5) for upadacitinib 15 mg and 30 mg, respectively. For HZ, the numbers were similar with IR 3.5 (95% 2.8, 4.2) and 6.2 (95% 5.0, 7.7) for upadacitinib 15 mg and 30 mg, respectively.

A phase III placebo-controlled RCT of upadacitinib 15 mg once daily for ankylosing spondylitis showed five (2.4%) individuals with serious infections (four COVID-19) in the upadacitinib group versus zero cases in the placebo group [91]. Furthermore, two cases (0.9% of individuals) of HZ versus zero cases in the control group were noted. Although this was a small study with relatively few individuals, these patients were younger than the typical RA population (mean age 42.4) with fewer comorbidities, further highlighting that upadacitinib may reduce viral infection resistance, as no cases were seen in the placebo group and HZ is normally less common in younger patients. An analysis of long-term safety data of upadacitinib across the different diagnoses (RA, PsA, AS, and AD) showed similar rates of serious infections across the different diagnoses, using 15 mg once daily, except in AD where 30 mg was also included. [92].

As with baricitinib, one study of upadacitinib in AD also reported a higher frequency of oral herpes simplex, with rates of 3% for 15 mg once daily and 8% for 30 mg once daily for upadacitinib-treated patients vs. 2% in the placebo group [93].

### 4.5. Filgotinib (Jyseleca)

Filgotinib is a selective inhibitor of JAK1, used for the treatment of RA and UC.

Combined clinical trial data showed a similar pattern to other JAKis [94]. Calculated IR for serious infections/100 PY was 2.2 (95% 1.8, 2.7) for filgotinib 100 mg and 1.9 (95% CI 1.6, 2.2) for 200 mg. The most common infections were COVID-19, pneumonia, cellulitis, and pyelonephritis. For HZ, the IR was 1.1 (95% 0.8, 1.5) for the 100 mg dose and 1.5 (95% CI 1.2, 1.8) for the 200 mg dose.

### 4.6. Ruxolitinib (Jakavi)

Ruxolitinib was developed as a JAK2 inhibitor but also affects JAK1. Unlike the JAKis described above, ruxolitinib is not used to treat inflammatory disease but instead is used for myeloproliferative diseases such as myelofibrosis, polycythemia vera, and essential thrombocytosis, as well as graft-versus-host-disease. It was the very first JAKi approved for use in humans. Infections as an adverse outcome have not been as extensively studied as with the other JAKis. One review of available data suggests limited evidence of an increased risk of serious infections compared to standard therapies. However, there was a clear elevated risk of HZ with ruxolitinib treatment, in doses ranging from 10 mg bid to 25 mg bid, as seen with other JAKis [95].

### 4.7. Abrocitinib (Cibinqo)

Abrocitinib is a selective JAK1 inhibitor approved for the treatment of AD. Combined data from clinical trials showed a similar incidence of serious infections between abrocitinib 100 mg and 200 mg groups and the placebo group [96]. A higher number of HZ cases was found in both abrocitinib groups, three cases (IR 1.90 (95% CI 0.39, 5.55)) in the 100 mg group and eight cases (IR 5.16 (95% 2.23, 10.16)) in the 200 mg group compared to zero cases in the placebo group. HSV was reported and with the incidence clearly increased in a dose-dependent manner: IR of 7.2 (95% CI 2.6, 14.58) in the placebo group, 12.07 (95% CI 7.26, 18.59) in the 100 mg group, and 16.22 (95% CI 10.49, 23.67) in the 200 mg group. A similar dose-dependent pattern was observed in a longer follow-up, though there were no longer any placebo controls. This trend was also seen in a smaller real-world descriptive analysis of 103 patients [97].

### 4.8. Ritlecitinib (Litfulo)

Ritlecitinib is a novel drug for the treatment of alopecia areata. It is a selective JAK3-inhibitor that also inhibits the TEC tyrosine kinase family, which have important roles in T-cell development. A safety analysis of the clinical trial program for ritlecitinib, with a combined median exposure of 624 days, using doses between 10 mg and 50 mg once daily with or without a 4-week loading dose of 200 mg once daily, showed an acceptable risk profile including a generally low rate of infections, with a serious infection IR of IR: 0.64/100 PY (95% CI 0.36, 1.06) [98]. Herpes zoster was more frequent in the ritlecitinib-treated individuals with an IR of 0.92 (95% CI 0.57, 1.40). HZ cases were generally non-serious, and most did not lead to permanent discontinuation. HSV infection rates were slightly higher in the ritlecitinib-treated individuals compared to placebo. No real-world data are yet available for safety regarding infections.

### 4.9. Delgotinib (Anzupgo)

Delgotinib is a pan-JAKi, exerting effect on all four JAKs, and is a used as a topical treatment for chronic hand eczema and in Japan for paediatric AD. A long-term safety analysis of an open-label clinical trial with a delgotinib cream of 20 mg/g twice daily over 36 weeks for individuals with chronic hand eczema showed a favourable safety profile regarding infections, with a case of eczema herpeticum following oral HSV infection in a patient with AD being among the only possibly drug related events [99]. Clinical trial data from Japan indicates a similarly satisfactory safety profile [100].

### 4.10. Deucravacitinib (Sotyktu)

Deucravacitinib is a TYK2 inhibitor used for the treatment of plaque psoriasis with trials ongoing for treatment of other conditions including Sjögrens disease and systemic lupus erythematosus. Combined safety data from the clinical trial program, including long-term extension, included 1519 patients receiving 6 mg of deucravacitinib once daily, for a cumulative exposure time of 4393 PY [101]. Analysis showed an exposure-adjusted IR of 1.99 (95% CI 1.59–2.46) for all serious infections, an exposure-adjusted IR of 0.80 (0.56–1.11) for serious infections excluding COVID-19, and a relatively low exposure-adjusted IR of 0.55 (0.35–0.82) for HZ. A small study of real-world data from Japan reported no infectious adverse effects; however, only 33 patients were included [102].

### 4.11. Fedratinib (Inrebic)

Fedratinib is a selective JAK2 inhibitor for the treatment of myeloproliferative disease. During clinical trials, using 400 mg once daily, few infectious complications were reported, with urinary tract infections being the only infectious adverse event, none of which were serious (graded 3 or higher out of 5) [103]. Data on infections for this drug is, however, very limited.

### 4.12. Tuberculosis

Risk of tuberculosis with JAKis is generally low, especially in areas with low TB prevalence, and is not higher than the risk observed with bDMARDs [104]. Screening for latent infection is recommended, and treatment should be initiated before starting JAKi treatment [105]. In a phase III studies of tofacitinib, no cases of reactivated TB were found among 263 patients identified with latent TB who started tofacitinib after treatment for latent infection [105].

### 4.13. Hepatitis B

Reactivation of latent hepatitis B is a known risk in patients treated with bDMARDs. Cases of reactivation have been reported for several JAKis including tofacitinib, baricitinib, and upadacitinib. In a retrospective Taiwanese study including 98 patients treated with tofacitinib, 8 patients were positive for hepatitis B surface antigen, indicating a presence of the virus [106]. Six out of these eight patients did not receive antiviral therapy and two experienced reactivation of hepatitis B infection during treatment with tofacitinib. Out of patients with detectable hepatitis B core antibodies, indicating prior contact with the virus but who did not have detectable antigen, only 2 out of 64 patients experienced reactivation. Recommendations in clinical practice is to screen for active or prior hepatitis B infection and, in positive cases, to consider antiviral prophylaxis and/or active monitoring [107].

### 4.14. Herpes Simplex Virus

Limited data are available on herpes infections other than herpes zoster in patients with RA. In dermatological disease, a meta-analysis of the use of JAKis for AA, AD, and PsA showed no significantly increased risk of herpes simplex infection across all diagnosis [108]. Abrocitinib and baricitinib were the most frequently used JAKi. However, for AD, which inherently carries an increased risk of herpes simplex infection, an additional, significant increase in risk was noted between individuals treated with JAKi versus placebo. No difference was seen between different classes of JAKi. Regarding AD, Tsai et al. showed an increased risk of HSV infection between JAKis (upadacitinib, abrocitinib, and baricitinib) and the biologic dupilumab [109]. The authors speculate that the difference might in part be attributed to the beneficial effect on AD by dupilumab but advises awareness regarding HSV in patients with AD treated with JAKi.

### 4.15. Lessons from COVID-19

The dramatic emergence of the COVID-19 pandemic resulted in a quick response to tackle the urgent need for new therapies. A hyperinflammatory condition affecting some individuals, which was associated with a worsened outcome, was identified and described in part with the model of the “cytokine storm” [110]. It shared several characteristics of the dysregulation in sepsis and toxic shock, with a massive release of pro-inflammatory cytokines, over-activation of lymphocytes, together with lymphopenia in severe cases [111,112]. Early on, corticosteroids were tried and confirmed to be beneficial in severe cases, particularly in those patients requiring oxygen or respiratory support [113]. The key pro-inflammatory cytokine IL-6 was targeted with monoclonal antibodies, tocilizumab, and/or sarilumab. Large, randomized studies showed a beneficial effect on mortality in the anti-IL-6 groups of patients with severe disease requiring organ support [114] or in those having hypoxia and signs of systemic inflammation measured by C-reactive protein levels [115].

With the plethora of cytokines involved, the idea of JAK inhibition was proposed. The JAKi baricitinib showed potential in disrupting the pathway of several of the cytokines involved in the inflammatory process. Baricitinib also has a unique affinity for AK1 adaptor protein–associated kinase 1 (AKK1) and cyclin G–associated kinase (GAK), kinases important for the endocytosis of angiotensin-converting enzyme 2 (ACE2). The binding of ACE2 by spike protein 1 in the alveolar epithelium and subsequent entry into the host cell by endocytosis is an important route of infection for the SARS-CoV-2 virus [116]. Several RCTs have confirmed the benefit of adding baricitinib to standard care in severe COVID-19 infections, showing significant improvement in mortality, clinical status, and reduced recovery time, without safety warnings [117,118]. Baricitinib is now a commonly recommended in COVID-19 treatment guidelines, often used in combination with steroids in hospitalized patients with worsening condition and signs of inflammatory activation.

Despite lacking the added effect of limiting ACE2-mediated endocytosis, tofacitinib has also been studied in COVID-19 considering its effect on the cytokine profile, although less rigorously than baricitinib. One small, randomized study from Italy showed that early treatment with tofacitinib gave a significant decrease in progression to mechanical ventilation (invasive or non-invasive) without safety concerns [119]. Another blinded RCT conducted in Brazil included 289 patients given tofacitinib in addition to standard care, most of which included dexamethasone [120]. This study showed a significant positive primary outcome of less death or respiratory failure through 28 days in patients receiving tofacitinib compared to placebo (RR 0.63 (p5% 0.41, 0.97)). For death alone, the difference was numerically but not statistically different, though the numbers were low. Overall, while there might be a distinct JAK effect of the JAKis in COVID-19, most of the evidence available supports baricitinib with its extra mode of action. As a result, tofacitinib is not recommended by the WHO for use in clinical practice, unless baricitinib and tocilizumab are unavailable [121].

## 5. Future Perspectives

Here, we have explored different aspects of manipulating the host’s immune system in infectious settings. While results highlight the dual nature of immune suppression with both harmful and beneficial effects depending on the infectious model, several key questions remain unanswered. Future research may aim to clarify the precise immunological mechanisms that dictate these opposing outcomes and explore potential strategies to mitigate the risks associated with immune modulation.

One key area for further studies is the clinical relevance of these findings. While animal models have provided valuable insights, translating these results into patient care remains challenging due to possible differences in immune response between species. Large-scale observational studies are needed to determine whether JAK inhibitors influence severity and outcomes of various infections in humans, particularly in patients with autoimmune diseases who are at baseline risk for infections. Given that infections like herpes zoster are well-documented adverse effects of tofacitinib treatment, prospective studies should assess whether certain patient populations are more vulnerable and whether preventive measures, such as vaccination strategies, can mitigate these risks.

Another important aspect is the interplay between JAK inhibition and cytokine-driven pathologies in infection. While JAK inhibitors have been shown to impair viral clearance in herpes virus encephalitis, their ability to reduce cytokine storms in severe infections, such as COVID-19, highlights their potential therapeutic role in hyperinflammatory states. Future studies may focus on identifying specific infectious scenarios where JAK inhibitors may confer a survival benefit versus those where they pose an unacceptable risk.

Furthermore, more precise immunomodulatory approaches should be explored. Rather than broadly inhibiting multiple cytokines, future therapies could aim for selective modulation of specific pathways involved in pathogenic inflammation while preserving antimicrobial immune responses. The development of next-generation JAK inhibitors with refined selectivity profiles or combination therapies that include antibiotics could offer a more balanced approach to treatment.

In summary, the findings discussed in this review highlight the complexity of immune suppression in infectious settings. While JAK inhibition can exacerbate certain infections, it may also offer protective benefits in others. A deeper mechanistic understanding, coupled with rigorous patient data, will be essential to optimizing the use of JAK inhibitors in patients at risk for infections.

## Figures and Tables

**Figure 1 pathogens-14-00324-f001:**
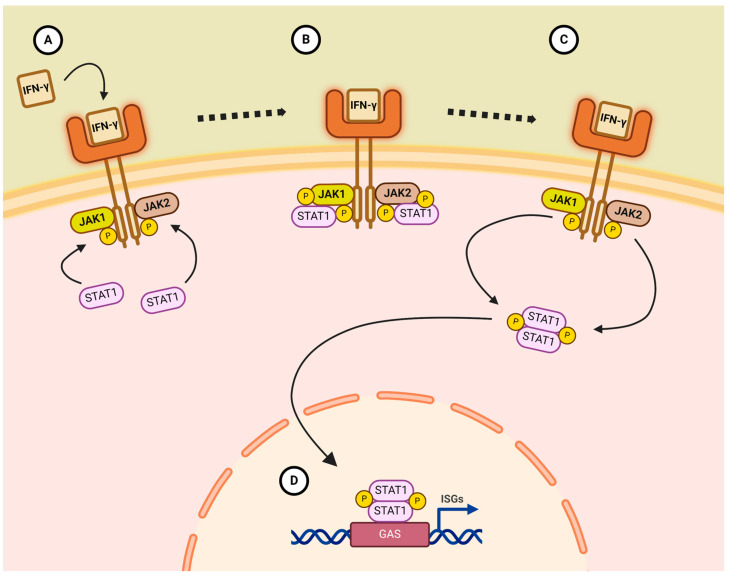
Schematic image of Janus kinase (JAK) signalling, exemplified by interferon-ɣ (IFN-γ). (**A**) Cytokine binds to its receptor allowing JAK1 and JAK2 to phosphorylate the Src homology 2 (SH2)-domain. This enables the binding of signal transducer and activator of transcription proteins (STATs). (**B**) The bound STATs are phosphorylated by the JAKs. (**C**) Phosphorylated STATs dimerize and translocate to the nucleus. (**D**) The STAT-dimer binds to gamma interferon activation site (GAS) and promotes expression of interferon stimulated genes (ISGs).

**Figure 2 pathogens-14-00324-f002:**
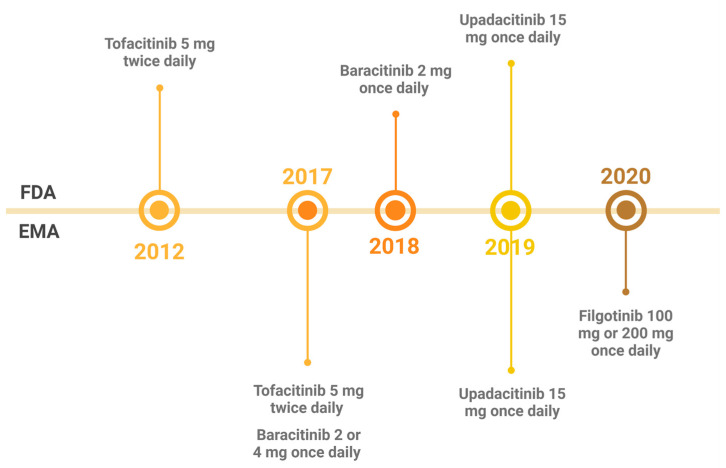
Timeline for JAKis approved for the treatment of rheumatoid arthritis by the Food and Drug Administration (FDA) (top) and the European Medicines Agency (EMA) (bottom).

**Table 2 pathogens-14-00324-t002:** Selection of JAKis in clinical use.

Drug	Target	Indication
Tofacitinib (Xeljans^®^)	JAK3, JAK1, JAK2	RA, UC, PsA, AS, JIA
Baricitinib (Olumiant^®^)	JAK1, JAK2	RA, AD, AA
Upadacitinib (Rinvoq^®^)	JAK1	RA, PsA, AS, nr-axSpa, AD, UC, CD
Filgotinib (Jyseleca^®^)	JAK1	RA, UC
Ruxolitinib (Jakavi^®^)	JAK1, JAK2	MPN, GVHD
Abrocitinib (Cibinqo^®^)	JAK1	AD
Deucravacitinib (Sotyktu^®^)	TYK2	PsO
Fedratinib (Inrebic^®^)	JAK2	MPN
Ritlecitinib (Litfulo^®^)	JAK3	AA
Delgotinib (Anzupgo^®^)	All 4 JAKs	AD

Abbreviations: RA rheumatoid arthritis, UC ulcerative colitis, PsA psoriatic arthritis, AS ankylosing spondylitis, nr-axSpA non-radiographic axial spondyloarthritis, JIA juvenile idiopathic arthritis, AD atopic dermatitis, AA alopecia areata, CD Crohn’s disease, MPN myeloproliferative neoplasm, GVHD graft-versus-host disease, PsO psoriasis.

**Table 3 pathogens-14-00324-t003:** The clinical outcomes of JAK/STAT component deficiency.

Infection Models in Genetically Modified Mice		Survival	Bacterial Load	Ref.
CLP model	STAT1−/−	+	−	[45]
	TYK2−/−	+/−	+/−	[45]
	STAT3−/−	−	+/−	[46]
	STAT4−/−	+	+/−	[47]
	STAT6−/−	+	−	[47]
LPS−induced shock	STAT1−/−	+	NA	[48]
	STAT2−/−	−	NA	[49]
	STAT3−/−	−	NA	[50]
	STAT4−/−	*	NA	[48,51]
	TYK2−/−	+	NA	[48,49]
	STAT6−/−	−	NA	[51]

+ = increase, − = decrease, +/− = no changes, NA = Not assessed, * = conflicting results.

**Table 4 pathogens-14-00324-t004:** The clinical outcomes of JAK/STAT inhibition in infection models.

Infection Models in JAK/STAT Inhibition		Survival	Bacterial/Viral Load	Ref.
CLP model	STAT3 inhibition (Rapamycin)	+	NA	[58]
	AG490 (JAK2 inhibition)	+	NA	[58,59]
	Tofacitinib (JAK3, JAK1, JAK2 inhibition)	+and less organ damage.	NA	[60]
LPS shock model	Ruxolitinib (JAK1/JAK2)	+	NA	[61]
	AG490	+	NA	[59]
	Tofacitinib	Less organ damage.	NA	[62]
Candida sepsis model	Ruxolitinib	Dose-dependent effect.	Dose dep.	[63]
*S. aureus* septic arthritis	Tofacitinib	More severe bone erosion.	+/−	[64]
*S. aureus* sepsis	Tofacitinib	+	NA	[64]
*S. aureus* enterotoxin-induced shock model	Tofacitinib	+	NA	[64,65]
HSV-1 encephalitis model	Tofacitinib	−	+	[66]

+ = increase, − = decrease, +/− = no changes, NA = Not assessed.

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
