# Peer review of "The Dual-Edged Sword: Risks and Benefits of JAK Inhibitors in Infections"

_pathogens, 2025, doi:10.3390/pathogens14040324_

Round 1

Reviewer 1 Report

Comments and Suggestions for Authors

The authors reviewed JAK inhibitors. Some of the findings may be useful, particularly the focus on infectious diseases.
A few questions.
(1) Ritlecitinib Tosilate and delgocitinib are not addressed. Please add them.
(2) It is known that diseases such as atopic dermatitis themselves increase the risk of herpes. The appropriateness regarding the use of JAK inhibitors in such a situation should be appropriately discussed.

Reviewer 2 Report

Comments and Suggestions for Authors

Journal

Pathogens (ISSN 2076-0817)

Manuscript ID:

pathogens-3526413

Type:

Review

Title:

THE DUAL-EDGED SWORD: RISKS AND BENEFITS OF JAK INHIBITORS IN INFECTIONS

Dear Editor,

Thank you for inviting me to review this manuscript submitted to Pathogens.

OVERALL COMMENTS

        Based on the statement that Janus kinase inhibitors (JAKi) are a new class of immunomodulatory drugs that have revolutionized the treatment landscape for autoimmune diseases. On the other hand, their ability to modulate immune responses presents a dual-edged nature, influencing both protective immunity and pathological inflammation. Due to these reasons, the authors intended to investigate the role of JAKi in infectious settings, highlighting both beneficial and detrimental effects. Their results showed that JAKi have a role in modulating hyperinflammatory responses in select infectious contexts. While they increase the risk of certain infections, they also show potential in managing hyperinflammatory conditions such as cytokine storms. “The key challenge is determining which patients and situations benefit most from JAKi while minimizing risks, requiring a careful and personalized treatment approach”.

        This is an interesting subject, but more than that, it is a very relevant subject in addressing different diseases that are increasing in prevalence throughout the world.

A small revision of the English needs to be made. There are small errors throughout the text.

_______

TITLE

        The title is adequate.

_______

ABSTRACT

        This section is adequate in content.

_______

KEYWORDS

        The included keywords are fine.

_______

  • INTRODUCTION

From lines 39-55, we can find a series of notes that are widely known. However, it is still necessary to include references, mainly published between 2023-2025.

_______

Section 2 is complete. Tables 1 and 2 are fine. I also liked Figure 1. Please include all the definitions of the abbreviations included in this Figure.

Sections 3 and 4 were also performed.

The section “4.9 Lessons from COVID-19” is very interesting.

  • CONCLUSION

I suggest expanding this section to “future perspectives”.

        Moreover, I suggest the inclusion of strengths and limitations of this review and how it can contribute to science in terms of using JAK inhibitors for known inflammatory and infectious diseases.

_______

  • REFERENCES

        I believe it is valuable to include newer references (Introduction and Discussion sections). Please include more references published in 2024 and also references published in 2025. PUBMED and EMBASE databases possess precious studies regarding JAK inhibitors.

I wish the authors good luck with the publication of this interesting review.

Comments on the Quality of English Language

Some minor corrections are necessary.

Reviewer 3 Report

Comments and Suggestions for Authors

In the current review the authors explored the complex role of janus kinase inhibitors in infectious settings, highlighting both beneficial and detrimental effects.

Some suggestions:

1.At the end of the introduction add please in which databases the literature search was conducted and in which period.

  1. page 2, line 69-70 – please add which are the “several other conditions” for which tofacitinib is used and its formulation.
  2. In my opinion more details concerning “JAK: Mechanism of action” would be welcome.
  3. Give please some more details concerning the infection models.

5.For all the studies on mice – add please:-  the type of mice used - the amount of the drug administered and the route/duration of administration (the information is given only for a part of the studies) 

  1. Point 3.2 Detrimental effects of JAKi in infection models:

You wrote only about tofacitinib. What about other JAK inhibitors?

7.Point 4 4. JAK-inhibitors and human infectious disease:

Add please the amount for all the administered drugs and the route/duration of administration.

  1. The information presented at point 4.2 Other JAKi for treatment of RA must be presented at points 4.3 and 4.4 were you presented Baricitinib (Olumiant) and Upadacitinib (Rinvoq). Filgotinib must be presented in a new subsection.
  2. No information are presented related to Deucravacitinib (Sotyktu) and Fedratinib (Inrebic). Please add about them before point 4.7. Tuberculosis.
  1. Please add a comparative discussion regarding the effectiveness and risk for the drugs presented in table 2.

  1. You wrote “The key challenge is determining which patients and situations benefit most from JAKi while minimizing risks, requiring a careful and personalized treatment approach”. At the end of the review, please add how you propose to achieve this.

  1. You forgot to add the authors contributions.

Reviewer 4 Report

Comments and Suggestions for Authors

The review is well organized and comprehensively described, scientifically sound and good English in general, but the work is presented as a general overview of jak inhibitors, but is essentially focused on Tofacitinib especially in the first part

Line 2: I suggest considering changing the title of the article, emphasizing that the object of study is essential Tofacinib and only marginally other jaki

Line 159: I suggest changing 3.1 "Beneficial impact of Jaki in infection models" to "Impact of Jaki in infection models", as not all JAKi show a beneficial impact.

Line 218: Amplify the first part and talk more extensively about other Jaki in infection models, if literature data is available. This part is essentially centered on Tofaccitinib only

Line 321 - comment similar to that of line 218.

Round 2

Reviewer 1 Report

Comments and Suggestions for Authors

none